**Data Availability Statement:** The datasets generated and/or analyzed during the current study are available in the DHS Program repository at

# Prevalence and correlates of tetanus toxoid uptake among women in sub-Saharan Africa: Multilevel analysis of demographic and health survey data

Richard Gyan Aboagye[1], Hubert Amu[2], Robert Kokou Dowou[3]*, Promise Bansah[3], Ijeoma Omosede Oaikhena[4], Luchuo Engelbert Bain[5,6]

1 Department of Family and Community Health, Fred N. Binka School of Public Health, University of Health and Allied Sciences, Hohoe, Ghana, 2 Department of Population and Behavioural Sciences, Fred N. Binka School of Public Health, University of Health and Allied Science, Hohoe, Ghana, 3 Department of Epidemiology and Biostatistics, Fred N. Binka School of Public Health, University of Health and Allied Science, Hohoe, Ghana, 4 Sault College Bachelors of Science in Nursing Program in Collaboration with Laurentian University, Sault Ste. Marie, Canada, 5 Department of Psychology, Faculty of Humanities, University of Johannesburg, Auckland Park, Johannesburg, South Africa, 6 International Research Development Centre, IDRC, Ottawa, Canada

* 2017rdowou@uhas.edu.gh

## Abstract

### Background

Tetanus toxoid vaccination is one of the most effective and protective measures against tetanus deaths among mothers and their newborns. We examined the prevalence and correlates of tetanus toxoid uptake among women in sub-Saharan African (SSA).

### Materials and methods

We analysed pooled data from the Demographic and Health Surveys (DHS) of 32 countries in SSA conducted from 2010 to 2020. We included 223,594 women with a history of childbirth before the survey. Percentages were used to present the prevalence of tetanus toxoid vaccine uptake among the women. We examined the correlates of tetanus toxoid uptake using a multilevel binary logistic regression.

### Results

The overall prevalence of tetanus toxoid uptake was 51.5%, which ranged from 27.5% in Zambia to 79.2% in Liberia. Women age, education level, current working status, parity, antenatal care visits, mass media exposure, wealth index, and place of residence were the factors associated with the uptake of tetanus toxoid among the women.

### Conclusion

Uptake of tetanus toxoid vaccination among the women in SSA was low. Maternal age, education, current working status, parity, antenatal care visits, exposure to mass media, and

https://dhsprogram.com/data/availabledatasets.cfm.

**Funding:** The author(s) received no specific funding for this work.

**Competing interests:** The authors have declared that no competing interests exist.

wealth status influence tetanus toxoid uptake among women. Our findings suggest that health sector stakeholders in SSA must implement interventions that encourage pregnant women to have at least four antenatal care visits. Also, health policymakers in SSA could ensure that the tetanus toxoid vaccine is free or covered under national health insurance to make it easier for women from poorer households to have access to it when necessary.

## Background

Globally, maternal and neonatal tetanus is still a substantial but preventable cause of mortality [1]. Tetanus disease is a life-threatening nervous system infection that is caused by the anaerobic bacterium Clostridium tetani [1]. The disease is known to affect all age groups; however, it is more prevalent and severe among neonates and pregnant women who have not been sufficiently immunized with tetanus-toxoid-containing vaccines as well as areas with poor hygiene [1, 2]. Tetanus continues to be a substantial cause of neonatal and maternal mortality in many sub-Saharan African (SSA) countries [1].

Neonatal tetanus happens during the first 28 days of life and maternal tetanus occurs during or within the first 6 weeks after pregnancy [3]. Evidence from the literature showed that an estimated 73,000 cases of tetanus including over 27,000 neonatal tetanus infections were recorded globally, among which an estimated 34,700 tetanus deaths occurred, with the largest burden in South Asia and SSA [4, 5]. Recent report from the World Health Organization (WHO) indicated that an about 25 000 newborns died from neonatal tetanus in 2018, a 97% reduction from 1988 when an estimated 787 000 newborn babies died of tetanus within their first month of life [6].

Tetanus toxoid vaccination is one of the most effective and protective measures against tetanus deaths among mothers and their newborns. As a result, it forms an indispensable component of antenatal care [7]. The initial global agenda of tetanus elimination by 2005 set by WHO was not achieved since maternal and neonatal mortality from tetanus remains high [8]. Globally, more than 79 million women and their babies remain unprotected against tetanus, leaving them at risk of maternal and neonatal tetanus infection and death [5]. Tetanus toxoid is a low-cost vaccine that is given at any time during pregnancy to protect women of childbearing age and newborns from tetanus during delivery [3, 9]. It was estimated that antenatal tetanus vaccination can reduce neonatal mortality by 94% if the majority of childbearing age women are immunized [7].

Per WHO's expanded program immunization recommendations, the first dose of tetanus toxoid vaccine is provided to women at first contact with healthcare services, and the second dose is provided 4 weeks later and at least 2 weeks before delivery [10–12]. Although the third dose should be given at least 6 months after the second, the last two boosters can be given during succeeding pregnancies or at least 1 year later [7, 13, 14]. A fully vaccinated pregnant woman could pass the acquired antibodies through the placenta to the fetus, thus protecting against tetanus until the newborn can be vaccinated at 6 weeks of age [14]. Cases of tetanus were less prevalent among newborns from women who received at least one dose of tetanus toxoid [2]. Case fatality rates from tetanus infection in resource-limited settings can be up to 100%, though with adequate medical care, it can be reduced to 10–20% [15]. Poor access to tetanus toxoid vaccines, lack of knowledge of women and misconceptions of vaccines as contraceptive agents are among the primary factors influencing tetanus toxoid coverage in SSA [16, 17].

This study examines the prevalence and correlates of tetanus toxoid uptake among women in 32 sub-Saharan African countries. Our study aims to provide a broader perspective on tetanus toxoid uptake among women in SSA as well as stimulate policy formulation and intervention development to improve its uptake in the sub-region.

## Materials and methods

We adopted the Strengthening the Reporting of Observational Studies in Epidemiology (STROBE) guidelines in drafting this paper [18].

### Data source and study design

We analysed pooled data from the Demographic and Health Survey (DHS) of 32 countries in SSA conducted from 2010 to 2020. Only 32 countries had data on all the variables of interest included in the study. The dataset used is freely available to download at https://dhsprogram.com/data/available-datasets.cfm. DHS is a comparable nationally representative survey undertaken regularly in over 90 countries, enhancing a global understanding of developing country health and demographic trends [19]. DHS employed a descriptive cross-sectional design to collect data from the respondents: men and women. Structured questionnaires were used to collect data from the respondents on several health and social indicators including uptake of tetanus toxoid [19, 20]. DHS utilised a two-stage cluster sampling method. First, a stratified sample of enumeration areas (EAs) was chosen using probability proportional to size (PPS). A listing technique was used in the designated EAs to ensure that all dwellings/households were listed. Second, households in the selected EAs were selected using equal probability systematic sampling, with the detailed sampling methodology highlighted in the literature [21]. In this study, we included 223,594 women with a history of childbirth before the survey.

### Variables

Andersen and Newman's Health Care Utilization model underpinned the selection of the variables for the study [22]. The model was first proposed in the 1960s [23]. The model is recognized as one of the analytical behavioral models that have been put forth to examine factors that influence the use of health services [22]. The model states that three factors—predisposing, enabling, and need for care—are what determine whether or not a person uses health services [22, 24]. Predisposing factors include social structural elements, demographic characteristics, and an individual's fundamental attitudes, beliefs, and knowledge toward health services [24]. Enabling factors consist of the available resources, both individually and in the community as well as avenues that facilitate a person to access health care [24]. On the other hand, the need for care factors includes the specific health problem, condition, illness, or preventive measure. The pregnant women's risk of being infected with tetanus served as the need for care factor per the Andersen and Newman's model. The individual characteristics, attitude, and practice of health behaviours, individual barriers to access health care, and resources constituted the individual level variables, whilst those around the women and in the households and community where those women reside were grouped as contextual level variables in the study.

Uptake of tetanus toxoid vaccination was the outcome variable in the study. To assess this variable, the women in DHS were asked whether they received tetanus toxoid vaccination during their last pregnancy. Those who responded receiving at least two doses based on the WHO's requirement were considered to receive tetanus toxoid vaccination and coded as 1 = yes (adequate), whilst the remaining women with less than two doses were grouped as 0 = no (inadequate). Other studies that utilised the DHS dataset employed similar categorization [25, 26].

We included sixteen explanatory variables in the study based on their influence from the literature [25–28] as well as their availability in the DHS dataset across the 32 countries. The variables were grouped into individual level and contextual levels. The individual level variables consisted of women's age, educational level, marital status, employment status, parity, read newspapers or magazines, watch television, listen to radio, health insurance coverage, number of antenatal care visits, getting medical help for self: permission to go, getting medical help for self: distance to the health facility, and getting medical help for self: getting money for treatment. On the other hand, household wealth index, place of residence, and geographic sub-regions were considered the contextual level variables. Detailed categorization of the variables has been shown in **Table 2**.

## Statistical analysis

We analysed the dataset using Stata version 17.0. We summarised the prevalence of tetanus toxoid uptake across the 32 countries using percentages. Next, we examine the distribution of tetanus toxoid uptake across the explanatory variables using cross-tabulations. Pearson chi-square test of independence was later used to determine the variables significantly associated with the uptake of tetanus toxoid. To ascertain the existence of collinearity among the studied variables, we conducted a multicollinearity test using the variance inflation factor (VIF). The minimum, maximum, and mean VIFs were 1.08, 4.56, and 2.18, respectively. Hence, there was no existence of high collinearity among the variables. We examined the factors associated with the uptake of tetanus toxoid using a multilevel binary logistic regression. Four (4) models were used. The first model (Model O) revealed the variance in tetanus toxoid uptake attributed to the primary sampling unit (PSU) by being an empty model with no explanatory variable. Model, I contained the individual-level variables, while Model II included contextual-level variables. Model III contained all the explanatory variables. The results were presented using adjusted odds ratios (aOR) with 95% confidence intervals (CIs). The "melogit" command in Stata was used to execute the multilevel regression models. To account for disproportionate sampling and non-response, the "svyset" command was used, and weighting was done to account for the intricate nature of DHS data.

## Ethical consideration

Ethical clearance was not sought for the present study due to the public availability of the DHS dataset. However, the DHS reported that ethical clearances were obtained from the Ethics Committee of ORC Macro Inc. as well as Ethics Boards of partner organizations of various countries such as the Ministries of Health. Written or oral consent from participant was not sought for this study because the study made use of secondary data from DHS dataset. The DHS follows the standards for ensuring the protection of respondents' privacy. All methods were carried out per relevant guidelines and regulations. Permission to download and use the dataset for publication purposes was sought from the MEASURE DHS and it was approved. We complied with all ethical guidelines regarding the use of secondary datasets for publication.

## Results

### Prevalence of tetanus toxoid uptake among women in sub-Saharan Africa

Table 1 presents the prevalence of tetanus toxoid uptake among women in SSA. The overall prevalence of tetanus toxoid uptake was 51.5% among women in SSA, which ranged from 27.5% in Zambia to 79.2% in Liberia.

**Table 1. Sample distribution and prevalence of tetanus toxoid uptake.**

| Country | Survey Year | Weighted N | Weighted % | Tetanus Toxoid uptake |
|---|---|---|---|---|
| 1. Angola | 2015–16 | 6,656 | 3.0 | 56.0 |
| 2. Burkina Faso | 2010 | 8,422 | 3.8 | 70.8 |
| 3. Benin | 2018 | 7,797 | 3.5 | 51.0 |
| 4. Burundi | 2016–17 | 8,693 | 3.9 | 28.5 |
| 5. Congo | 2011–12 | 4,763 | 2.1 | 60.6 |
| 6. DR Congo | 2013–14 | 8,996 | 4.0 | 43.3 |
| 7. Cote d'Ivoire | 2011–12 | 4,733 | 2.1 | 55.6 |
| 8. Cameroon | 2018 | 7,321 | 3.3 | 53.6 |
| 9. Ethiopia | 2016 | 8,068 | 3.6 | 41.1 |
| 10. Gabon | 2012 | 3,625 | 1.6 | 68.8 |
| 11. Ghana | 2014 | 4,420 | 2.0 | 57.1 |
| 12. Gambia | 2019–20 | 5,359 | 2.4 | 35.4 |
| 13. Guinea | 2018 | 5,261 | 2.3 | 48.1 |
| 14. Kenya | 2014 | 14,524 | 6.5 | 51.2 |
| 15. Comoros | 2012 | 2,648 | 1.2 | 35.0 |
| 16. Liberia | 2019–20 | 3,710 | 1.7 | 79.2 |
| 17. Lesotho | 2014 | 3,202 | 1.4 | 58.2 |
| 18. Mali | 2018 | 5,334 | 2.4 | 35.7 |
| 19. Malawi | 2015–16 | 12,035 | 5.4 | 73.0 |
| 20. Nigeria | 2018 | 20,502 | 9.2 | 52.9 |
| 21. Niger | 2012 | 5,680 | 2.5 | 49.7 |
| 22. Namibia | 2013 | 4,722 | 2.1 | 36.3 |
| 23. Rwanda | 2019–20 | 7,759 | 3.5 | 33.6 |
| 24. Sierra Leone | 2019 | 7,541 | 3.4 | 78.1 |
| 25. Senegal | 2010–11 | 7,209 | 3.2 | 57.4 |
| 26. Chad | 2014–15 | 8,707 | 3.9 | 51.0 |
| 27. Togo | 2013–14 | 4,471 | 2.0 | 62.0 |
| 28. Tanzania | 2015–16 | 6,494 | 2.9 | 51.8 |
| 29. Uganda | 2016 | 8,925 | 4.0 | 61.3 |
| 30. South Africa | 2018 | 4,195 | 1.9 | 28.8 |
| 31. Zambia | 2018 | 6,813 | 3.0 | 27.5 |
| 32. Zimbabwe | 2015 | 5,009 | 2.2 | 40.0 |
| **All countries** | **2010–2020** | **223,594** | **100.0** | **51.5** |

## Distribution of tetanus toxoid uptake across the explanatory variables

Table 2 presents the distribution of tetanus toxoid uptake across the explanatory variables. The proportion of tetanus toxoid uptake was highest among women aged 20–24 (56.4%), those with higher education (61.7%), and those from richest wealth index households (60.1%). In terms of marital status, while women who were separated had the highest tetanus toxoid uptake (56.3%), those women who were married had the lowest proportion (50.4%). Concerning current working status, we found that women who were working had the highest proportion (52.9%) of tetanus toxoid uptake. Tetanus toxoid uptake was prevalent among women with one birth history (60.6%) and those who had four or more antenatal care visits (61.1%). We found that women who were not covered by health insurance had a 52.1% uptake of tetanus toxoid. The results also showed that the proportion of tetanus toxoid uptake was high among women who had no problem in getting permission to go to a health facility (52.1%),

**Table 2. Distribution of tetanus toxoid uptake across the explanatory variables.**

| Variables | Weighted N | Weighted % | Tetanus toxoid uptake | | |
|---|---|---|---|---|---|
| | | | No (%) | Yes (%) | P-value |
| **Women's age (years)** | | | | | <0.001 |
| 15–19 | 16,005 | 7.2 | 45.0 | 55.0 | |
| 20–24 | 49,632 | 22.2 | 43.6 | 56.4 | |
| 25–29 | 57,700 | 25.8 | 47.1 | 52.9 | |
| 30–34 | 45,628 | 20.4 | 50.3 | 49.7 | |
| 35–39 | 33,167 | 14.8 | 53.2 | 46.8 | |
| 40–44 | 16,186 | 7.2 | 54.9 | 45.1 | |
| 45–49 | 5,276 | 2.4 | 54.7 | 45.3 | |
| **Level of education** | | | | | <0.001 |
| No education | 80,913 | 36.2 | 53.8 | 46.2 | |
| Primary | 73,089 | 32.7 | 48.4 | 51.6 | |
| Secondary | 60,400 | 27.0 | 42.9 | 57.1 | |
| Higher | 9,192 | 4.1 | 38.3 | 61.7 | |
| **Marital status** | | | | | <0.001 |
| Never married | 18,963 | 8.5 | 46.7 | 53.3 | |
| Married | 154,356 | 69.0 | 49.6 | 50.4 | |
| Cohabiting | 34,046 | 15.2 | 45.3 | 54.7 | |
| Widowed | 3,257 | 1.5 | 49.4 | 50.6 | |
| Divorced | 4,019 | 1.8 | 48.9 | 51.1 | |
| Separated | 8,953 | 4.0 | 43.7 | 56.3 | |
| **Current working status** | | | | | <0.001 |
| Not working | 82,414 | 36.9 | 50.7 | 49.3 | |
| Working | 141,180 | 63.1 | 47.1 | 52.9 | |
| **Parity** | | | | | <0.001 |
| One birth | 48,048 | 21.5 | 39.4 | 60.6 | |
| Two births | 42,848 | 19.2 | 46.2 | 53.8 | |
| Three births | 36,017 | 16.1 | 48.5 | 51.5 | |
| Four or more births | 96,681 | 43.2 | 54.0 | 46.0 | |
| **Number of antenatal care visits** | | | | | <0.001 |
| None | 22,015 | 9.8 | 92.9 | 7.1 | |
| 1–3 visits | 72,879 | 32.6 | 51.9 | 48.1 | |
| 4 or more visits | 128,700 | 57.6 | 38.9 | 61.1 | |
| **Covered by health insurance** | | | | | <0.001 |
| No | 202,232 | 90.4 | 47.9 | 52.1 | |
| Yes | 21,362 | 9.6 | 53.7 | 46.3 | |
| **Getting medical help for self: Permission to go** | | | | | <0.001 |
| Not a big problem | 181,268 | 81.1 | 47.9 | 52.1 | |
| Big problem | 42,326 | 18.9 | 51.0 | 49.0 | |
| **Getting medical help for self: Distance to health facility** | | | | | <0.001 |
| Not a big problem | 136,020 | 60.8 | 47.3 | 52.7 | |
| Big problem | 87,574 | 39.2 | 50.3 | 49.7 | |
| **Getting medical help for self: Getting money for treatment** | | | | | 0.002 |
| Not a big problem | 102,441 | 45.8 | 47.9 | 52.1 | |
| Big problem | 121,153 | 54.2 | 49.0 | 51.0 | |
| **Watch television** | | | | | <0.001 |
| No | 134,628 | 60.2 | 51.3 | 48.7 | |

*(Continued)*

**Table 2.** (Continued)

| Variables | Weighted N | Weighted % | Tetanus toxoid uptake | | |
|---|---|---|---|---|---|
| | | | No (%) | Yes (%) | P-value |
| Yes | 88,966 | 39.8 | 44.1 | 55.9 | |
| **Read newspaper or magazine** | | | | | <0.001 |
| No | 185,438 | 82.9 | 49.4 | 50.6 | |
| Yes | 38,156 | 17.1 | 43.7 | 56.3 | |
| **Listen to radio** | | | | | <0.001 |
| No | 95,949 | 42.9 | 52.2 | 47.8 | |
| Yes | 127,645 | 57.1 | 45.6 | 54.4 | |
| **Wealth index** | | | | | <0.001 |
| Poorest | 47,429 | 21.2 | 55.3 | 44.7 | |
| Poorer | 47,012 | 21.0 | 51.3 | 48.7 | |
| Middle | 44,879 | 20.1 | 48.4 | 51.6 | |
| Richer | 43,958 | 19.7 | 45.9 | 54.1 | |
| Richest | 40,315 | 18.0 | 39.9 | 60.1 | |
| **Place of residence** | | | | | <0.001 |
| Urban | 77,672 | 34.7 | 43.0 | 57.0 | |
| Rural | 145,922 | 65.3 | 51.3 | 48.7 | |
| **Sub-region** | | | | | <0.001 |
| Central | 40,067 | 19.9 | 46.7 | 53.3 | |
| Eastern | 57,112 | 25.5 | 55.2 | 44.8 | |
| Southern | 35,977 | 16.1 | 51.5 | 48.5 | |
| Western | 90,438 | 40.5 | 43.8 | 56.2 | |

*p-values were generated from chi-square test

distance to the health facility (52.7%), and getting money for treatment (52.1%). Tetanus toxoid uptake was prevalent among women who read newspapers or magazines (56.3%), listened to the radio (54.4%), watched television (55.9%), and those who resided in urban areas (57.0%).

## Factors associated with the uptake of tetanus toxoid vaccination among women in sub-Saharan Africa

Table 3, Model III presents the results of the factors associated with the uptake of tetanus toxoid vaccination among women in SSA. Women aged 20–24 (AOR = 1.17, 95%CI = 1.11, 1.24), 25–29 (AOR = 1.15, 95%CI = 1.08, 1.23), and 30–34 (AOR = 1.08, 95%CI = 1.01, 1.15) were more likely to receive adequate tetanus toxoid vaccination compared to those aged 15–19. Women with primary (AOR = 1.10, 95%CI = 1.06, 1.14) and higher education levels (AOR = 1.10, 95% CI = 1.01, 1.20) had higher odds of adequate tetanus toxoid uptake compared to those with no formal education. Compared to women who had never been in union, those who were married, cohabiting, divorced, widowed, and separated were more likely to receive adequate tetanus toxoid vaccination. Women who were currently working had higher odds (AOR = 1.06, 95% CI = 1.03, 1.10) of tetanus toxoid uptake compared with those who were not working.

Concerning parity, the odds of tetanus toxoid uptake decreased with an increase in the number of births with women who had four or more births having the lowest odds (AOR = 0.59, 95% CI = 0.57, 0.63). Women who have had 1–3 visits (AOR = 14.42, 95%

**Table 3. Mixed-effect analysis of correlates of tetanus toxoid uptake among women in sub-Saharan Africa.**

| Variable | Model O | Model I AOR [95% CI] | Model II AOR [95% CI] | Model III AOR [95% CI] |
|---|---|---|---|---|
| *Fixed-effect results* | | | | |
| **Women's age (years)** | | | | |
| 15–19 | | 1.00 | | 1.00 |
| 20–24 | | 1.11*** [1.05, 1.18] | | 1.17*** [1.11, 1.24] |
| 25–29 | | 1.08* [1.01, 1.15] | | 1.15*** [1.08, 1.23] |
| 30–34 | | 0.99 [0.93, 1.06] | | 1.08* [1.01, 1.15] |
| 35–39 | | 0.92* [0.86, 0.99] | | 1.01 [0.94, 1.08] |
| 40–44 | | 0.90* [0.83, 0.98] | | 1.00 [0.92, 1.08] |
| 45–49 | | 0.97 [0.88, 1.07] | | 1.07 [0.97, 1.19] |
| **Level of education** | | | | |
| No education | | 1.00 | | 1.00 |
| Primary | | 0.88*** [0.85, 0.91] | | 1.10*** [1.06, 1.14] |
| Secondary | | 0.93*** [0.89, 0.97] | | 1.02 [0.98, 1.07] |
| Higher | | 1.07 [0.98, 1.17] | | 1.10* [1.01, 1.20] |
| **Marital status** | | | | |
| Never married | | 1.00 | | 1.00 |
| Married | | 1.25*** [1.18, 1.32] | | 1.14*** [1.08, 1.21] |
| Cohabiting | | 1.35*** [1.27, 1.44] | | 1.26*** [1.19, 1.34] |
| Widowed | | 1.37*** [1.22, 1.53] | | 1.37*** [1.22, 1.54] |
| Divorced | | 1.21*** [1.10, 1.35] | | 1.29*** [1.16, 1.43] |
| Separated | | 1.40*** [1.29, 1.52] | | 1.40*** [1.29, 1.52] |
| **Current working status** | | | | |
| Not working | | 1.00 | | 1.00 |
| Working | | 1.12*** [1.09, 1.15] | | 1.06*** [1.03, 1.10] |
| **Parity** | | | | |
| One birth | | 1.00 | | 1.00 |
| Two births | | 0.73*** [0.70, 0.76] | | 0.72*** [0.69, 0.75] |
| Three births | | 0.68*** [0.65, 0.72] | | 0.67*** [0.63, 0.70] |
| Four or more births | | 0.62*** [0.59, 0.66] | | 0.59*** [0.57, 0.63] |
| **Number of antenatal care visits** | | | | |
| None | | 1.00 | | 1.00 |
| 1–3 visits | | 13.19*** [11.87, 14.64] | | 14.42*** [12.92, 16.09] |
| 4 or more visits | | 22.65*** [20.39, 25.17] | | 24.03*** [21.54, 26.81] |
| **Covered by health insurance** | | | | |
| No | | 1.00 | | 1.00 |
| Yes | | 0.63*** [0.60, 0.66] | | 0.68*** [0.65, 0.71] |
| **Getting medical help for self: Getting money for treatment** | | | | |
| Not a big problem | | 1.00 | | 1.00 |
| Big problem | | 1.13*** [1.10, 1.16] | | 1.08*** [1.05, 1.12] |
| **Getting medical help for self: Distance to health facility** | | | | |
| Not a big problem | | 1.00 | | 1.00 |
| Big problem | | 1.02 [0.99, 1.06] | | 1.08*** [1.05, 1.12] |
| **Getting medical help for self: Permission to go** | | | | |
| Not a big problem | | 1.00 | | 1.00 |
| Big problem | | 1.06** [1.02, 1.11] | | 0.98 [0.94, 1.02] |
| **Read newspaper or magazine** | | | | |

*(Continued)*

**Table 3.** (Continued)

| Variable | Model O | Model I<br>AOR [95% CI] | Model II<br>AOR [95% CI] | Model III<br>AOR [95% CI] |
|---|---|---|---|---|
| No | | 1.00 | | 1.00 |
| Yes | | 0.97 [0.93, 1.01] | | 1.08*** [1.03, 1.13] |
| **Listen to radio** | | | | |
| No | | 1.00 | | 1.00 |
| Yes | | 1.07*** [1.04, 1.10] | | 1.07*** [1.04, 1.11] |
| **Watch television** | | | | |
| No | | 1.00 | | 1.00 |
| Yes | | 1.03 [1.00, 1.06] | | 0.87*** [0.84, 0.91] |
| **Wealth index** | | | | |
| Poorest | | | 1.00 | 1.00 |
| Poorer | | | 1.16*** [1.12, 1.20] | 1.05** [1.01, 1.10] |
| Middle | | | 1.28*** [1.23, 1.34] | 1.11*** [1.07, 1.16] |
| Richer | | | 1.39*** [1.32, 1.46] | 1.19*** [1.13, 1.26] |
| Richest | | | 1.76*** [1.67, 1.86] | 1.48*** [1.40, 1.57] |
| **Place of residence** | | | | |
| Urban | | | 1.00 | 1.00 |
| Rural | | | 0.95* [0.90, 0.99] | 1.15*** [1.10, 1.21] |
| **Sub-region** | | | | |
| Central | | | 1.00 | 1.00 |
| Eastern | | | 0.69*** [0.65, 0.73] | 0.56*** [0.52, 0.60] |
| Southern | | | 0.81*** [0.76, 0.86] | 0.54*** [0.50, 0.57] |
| Western | | | 1.12*** [1.06, 1.18] | 1.06 [0.99, 1.12] |
| **Random effect model** | | | | |
| PSU variance (95% CI) | 0.46 [0.39, 0.54] | 0.39 [0.33, 0.47] | 0.40 [0.34, 0.47] | 0.37 [0.32, 0.44] |
| ICC | 0.12 | 0.11 | 0.11 | 0.10 |
| Wald chi-square | Reference | 6224.39*** | 1295.33*** | 7087.55*** |
| **Model fitness** | | | | |
| Log-likelihood | -313288.98 | -284367.57 | -309156.47 | -280472.7 |
| AIC | 626582 | 568793.1 | 618332.9 | 561019.4 |
| N | 223594 | 223594 | 223594 | 223594 |
| Number of clusters | 1611 | 1611 | 1611 | 1611 |

aOR = adjusted odds ratios; CI = Confidence Interval

* $p < 0.05$

** $p < 0.01$

*** $p < 0.001$

1.00 = Reference category; PSU = Primary Sampling Unit; ICC = Intra-Class Correlation; AIC = Akaike Information Criterion

CI = 12.92, 16.09) and those with 4 or more antenatal care visits (AOR = 24.03, 95% CI = 21.54, 26.8) were more likely to receive adequate tetanus toxoid vaccination compared to those with no history of antenatal care visits. Lower odds of adequate tetanus toxoid vaccination uptake was found among women covered by health insurance (AOR = 0.68, 95% CI = 0.65, 0.71) and those who watched television (AOR = 0.87, 95%CI = 0.84, 0.91). Women who had problems in getting money for treatment (AOR = 1.08, 95%CI = 95% CI = 1.05, 1.12) and those who had problems in regarding distance to health facility (AOR = 1.08, 95% CI = 1.05, 1.12) were more likely to receive adequate tetanus toxoid vaccination.

Women who read newspapers or magazines (AOR = 1.08, 95% CI = 1.03, 1.13) and those who listened to radio (AOR = 1.07, 95% CI = 1.04, 1.11) had higher odds of tetanus toxoid uptake compared to those who did not. The odds of tetanus toxoid uptake increased with increasing wealth index with the highest odds among women from the richest wealth quintile households (AOR = 1.48, 95% CI = 1.40, 1.57). Women residing in rural areas (AOR = 1.15, 95% CI = 1.10, 1.21) were more likely to receive adequate tetanus toxoid vaccination relative to those in urban areas. At the sub-regional level, women from Eastern (AOR = 0.56, 95% CI = 0.52, 0.60) and Southern (AOR = 0.54, 95% CI = 0.50, 0.57) parts of SSA were less likely to receive adequate tetanus toxoid vaccination compared to those in the Central SSA.

## Discussion

We examined the prevalence and correlates of tetanus toxoid uptake among women in SSA using DHS data from 32 countries. We found that the overall prevalence of tetanus toxoid uptake was 51.5%. We, however, found country-level variations in the prevalence of tetanus toxoid uptake with the highest proportion being recorded in Liberia (79.2%) and the lowest in Zambia (27.5%). The low uptake recorded in most of the countries like Zambia for instance could be ascribed to numerous factors such as poor health education, non-availability and accessibility to health service in areas especially the rural settings, and other health system factors in these sub-Saharan African countries [29–32].

Women with primary and higher educational levels were more likely to receive adequate tetanus toxoid compared to those with no education. This finding is consistent with findings from previous studies that indicated that women with advanced level education are more willing to accept health interventions [33–37]. The higher likelihood of update among women with formal education could be attributed to the fact that those women are well informed and have better knowledge on the benefits of acceptance and the consequences of non-acceptance of tetanus toxoid for both themselves and their children [38, 39]. Also, women with formal education may well empowered socially and economically to afford the cost of health care services [36, 38–40].

We found that women who were currently working had higher odds of tetanus toxoid uptake compared with those who were not working. This finding corroborates with previous studies that found higher likelihood of tetanus toxoid vaccination uptake among women who were employed or working [17, 40, 41]. The financial earnings of working women could have empowered them with the purchasing power to afford transportation to the facility and cost of health services in situations where the services were not entirely free or not covered by health insurance [26, 40, 42–45].

Parity was a strong correlate of tetanus toxoid uptake. In this regard, we found that the odds of tetanus toxoid uptake decreases with an increase in number of births with women who had four or more births having the lowest odds. This finding is consistent with a previous study that made a similar argument on parity [11, 25, 28, 46, 47]. This finding of lesser odds of tetanus toxoid uptake among women with more than one birth in this study could be attributed to the preceding experience of women with pregnancy, delivery, and service provision by health workers, including the side effects of tetanus toxoid vaccination, which could have served as deterrent factors [28, 30, 46].

We found that the number of antenatal care visits correlates with tetanus toxoid uptake. Women who had at least one antenatal care visits were more likely to receive tetanus toxoid vaccination. This observation is consistent with previous studies that made similar observations of antenatal care attendance influencing tetanus toxoid uptake [17, 47–50]. Women who attend antenatal care services are more likely to have been educated on the importance of tetanus toxoid vaccination in protecting them and their unborn child from having tetanus. Also,

tetanus toxoid vaccination is one of the components of antenatal care services, therefore, health workers ensure that woman receives tetanus toxoid immunization [25, 46, 50].

Our study found that women who read newspapers or magazines and those who listened to radio were more likely to receive adequate tetanus toxoid vaccination compared to those who did not. This finding is congruent with previous studies that noted the influence of mass media on women's uptake of tetanus toxoid uptake [25, 42, 50–52]. Women who were exposed to mass media, except for watching television might have received information on the benefits of tetanus toxoid vaccine, which could have subsequently influenced their decision to take the vaccine [53]. Also, the women might have been educated about the importance of receiving the tetanus toxoid or the possible consequences of not receiving the vaccine for both the unborn child and themselves. Hence, their decision to receive the recommended doses of the tetanus toxoid vaccine [25, 38, 50, 54].

Our study also showed that the odds of receiving tetanus toxoid increased with wealth status with women from the richest household having the highest likelihood. Our result is consistent with findings from previous studies where wealthy women were more likely to receive tetanus toxoid vaccination [35, 45, 48, 54–56]. Women from wealthy households would have the financial capabilities to cater for cost associated with health services including preventive services such as uptake of tetanus toxoid [7, 17, 26, 57–59].

## Conclusion

Our study has shown that tetanus toxoid uptake among women in SSA was low. Factors identified to be associated with tetanus toxoid uptake were the age of the women, education, marital status, health insurance coverage, current working status, parity, antenatal care visits, mass media exposure, wealth status, and place of residence. Our findings suggest that health sector stakeholders in SSA must implement interventions that encourage pregnant women to have at least four antenatal care visits leveraging on platforms such as the radio and the print media. Also, policymakers in sub-Saharan African countries could ensure that tetanus toxoid vaccine is free or covered under national health insurance to make it easier for women from poorer households to have access to it when necessary.

## Acknowledgments

We would like to thank the DHS Program for making the data available for the study.

## Author Contributions

**Conceptualization:** Hubert Amu, Robert Kokou Dowou, Luchuo Engelbert Bain.

**Formal analysis:** Richard Gyan Aboagye, Robert Kokou Dowou, Promise Bansah.

**Methodology:** Richard Gyan Aboagye, Robert Kokou Dowou.

**Resources:** Promise Bansah.

**Supervision:** Hubert Amu, Robert Kokou Dowou, Luchuo Engelbert Bain.

**Validation:** Hubert Amu.

**Visualization:** Hubert Amu, Luchuo Engelbert Bain.

**Writing – original draft:** Robert Kokou Dowou, Ijeoma Omosede Oaikhena.

**Writing – review & editing:** Richard Gyan Aboagye, Hubert Amu, Robert Kokou Dowou, Promise Bansah, Ijeoma Omosede Oaikhena, Luchuo Engelbert Bain.

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
