## [Decision Letter · Decision Letter 0]

7 Sep 2023

PONE-D-23-03453Prevalence and correlates of tetanus toxoid uptake among women in sub-Saharan Africa: Multilevel analysis of demographic and health survey dataPLOS ONE

Dear Dr. Dowou,

Thank you for submitting your manuscript to PLOS ONE. After careful consideration, we feel that it has merit but does not fully meet PLOS ONE’s publication criteria as it currently stands. Therefore, we invite you to submit a revised version of the manuscript that addresses the points raised during the review process.

We look forward to receiving your revised manuscript.

Kind regards,

Frank T. Spradley

Academic Editor

PLOS ONE

- https://doi.org/10.1371/journal.pone.0276417

- https://tropmedhealth.biomedcentral.com/articles/10.1186/s41182-022-00398-4#citeas

In your revision ensure you cite all your sources (including your own works), and quote or rephrase any duplicated text outside the methods section. Further consideration is dependent on these concerns being addressed.

Reviewers' comments:

Reviewer's Responses to Questions

**Comments to the Author**

1. Is the manuscript technically sound, and do the data support the conclusions?

Reviewer #1: Partly

2. Has the statistical analysis been performed appropriately and rigorously? 

Reviewer #1: Yes

3. Have the authors made all data underlying the findings in their manuscript fully available?

Reviewer #1: Yes

4. Is the manuscript presented in an intelligible fashion and written in standard English?

Reviewer #1: Yes

5. Review Comments to the Author

Reviewer #1: This is a timely and insightful piece of work on the prevalence and correlates of tetanus toxoid vaccination in sub-Saharan Africa. It is clear, concise and outlines in detail the exact variables it set out to measure as well as the specific tests carried out. It also provides important findings that health policy makers can use in interventions. There are however a few corrections and suggestions for you to consider in making the paper more robust.

1. The variables: ‘health insurance coverage, antenatal care attendance, getting medical help for self: permission to go, getting medical help for self: distance to the health facility, and getting medical help for self: getting money for treatment’ could be added to the contextual variables instead of individual variables. They appear to be the context of the dependent variable.

2. The variable ‘sex of household head’ is listed as a contextual variable but was not included in the chi-square and multi-level analysis. Is there a reason why?

3. While the literature on tetanus immunization uptake have been highlighted, I suggest that you could make use of a theoretical background or framework which provides a logical relationship between the variables and also help establish apriori expectations. The framework would provide a solid rationale for the work. For instance the socio-ecological model can be useful. It will enable you look at the multiple levels of influence on health behaviors or variables and their outcomes. This can also help you delineate the individual level variables and contextual variables properly

4. Lastly, the paper will benefit from some proof reading. For instance in the discussion section ‘update’ should read ‘uptake’ and ‘them themselves’ should read ‘ themselves’ .

6. PLOS authors have the option to publish the peer review history of their article (what does this mean?). If published, this will include your full peer review and any attached files.

Reviewer #1: No

---

## [Author Response · Author response to Decision Letter 0]

1 Nov 2023

Reviewers’ comments

Reviewer #1: This is a timely and insightful piece of work on the prevalence and correlates of tetanus toxoid vaccination in sub-Saharan Africa. It is clear, concise and outlines in detail the exact variables it set out to measure as well as the specific tests carried out. It also provides important findings that health policy makers can use in interventions. There are however a few corrections and suggestions for you to consider in making the paper more robust.

Response: Thank you for making time to review our manuscript.

1. The variables: ‘health insurance coverage, antenatal care attendance, getting medical help for self: permission to go, getting medical help for self: distance to the health facility, and getting medical help for self: getting money for treatment’ could be added to the contextual variables instead of individual variables. They appear to be the context of the dependent variable.

Response: Thank you. These variables pertain to issues specific to the respondents (women) and are not contextual. Plethora of studies using the DHS dataset also confirms the inclusion of these variables at the individual level and not the contextual level.

2. The variable ‘sex of household head’ is listed as a contextual variable but was not included in the chi-square and multi-level analysis. Is there a reason why?

Response: This variable was dropped and not utilised throughout the study. Hence, its absence from the Tables. 

3. While the literature on tetanus immunization uptake have been highlighted, I suggest that you could make use of a theoretical background or framework which provides a logical relationship between the variables and also help establish apriori expectations. The framework would provide a solid rationale for the work. For instance the socio-ecological model can be useful. It will enable you look at the multiple levels of influence on health behaviors or variables and their outcomes. This can also help you delineate the individual level variables and contextual variables properly

Response: We have provided a theoretical model that informed the selection of the variables included in the study.

4. Lastly, the paper will benefit from some proof reading. For instance in the discussion section ‘update’ should read ‘uptake’ and ‘them themselves’ should read ‘ themselves’ .

Response: Thank you. We have thoroughly proofread the manuscript to correct grammatical and typographical errors.

---

## [Editor Report · Decision Letter 1]

4 Dec 2023

PONE-D-23-03453R1Prevalence and correlates of tetanus toxoid uptake among women in sub-Saharan Africa: Multilevel analysis of demographic and health survey dataPLOS ONE

Dear Dr. Dowou,

Thank you for submitting your manuscript to PLOS ONE. After careful consideration, we feel that it has merit but does not fully meet PLOS ONE’s publication criteria as it currently stands. Therefore, we invite you to submit a revised version of the manuscript that addresses the points raised during the review process.

*Specifically, it is not clear where in the resubmission the responses to reviewer comments were applied. Please submit a version of the marked-up copy (detail below in red text) of the manuscript tracking where changes were made according to suggestions from the previous round of reviews (the current document does not track changes, so it is not evident that any revisions were made). Replies to all comments must be marked in the revised manuscript, unless it is stated in the response-to-reviewers' comments document that no changes were made in the revised manuscript.*

We look forward to receiving your revised manuscript.

Kind regards,

Frank T. Spradley

Academic Editor

PLOS ONE

---

## [Author Response · Author response to Decision Letter 1]

5 Dec 2023

Authors are pleased to resubmit the revised manuscript for consideration, after a review by reviewers. We believe we have addressed the comments thoroughly. 

I present below a point-by-point of the concerns raised and how they have been addressed in this revised submission. The authors revised feedback appears in track changes and appears as the authors' response.

We look forward to your positive feedback on our submission.

Thank you

(Corresponding author)

Reviewers’ comments

Reviewer #1: This is a timely and insightful piece of work on the prevalence and correlates of tetanus toxoid vaccination in sub-Saharan Africa. It is clear, concise and outlines in detail the exact variables it set out to measure as well as the specific tests carried out. It also provides important findings that health policy makers can use in interventions. There are however a few corrections and suggestions for you to consider in making the paper more robust.

Response: Thank you for making time to review our manuscript.

1. The variables: ‘health insurance coverage, antenatal care attendance, getting medical help for self: permission to go, getting medical help for self: distance to the health facility, and getting medical help for self: getting money for treatment’ could be added to the contextual variables instead of individual variables. They appear to be the context of the dependent variable.

Response: Thank you. These variables pertain to issues specific to the respondents (women) and are not contextual. Plethora of studies using the DHS dataset also confirms the inclusion of these variables at the individual level and not the contextual level.

2. The variable ‘sex of household head’ is listed as a contextual variable but was not included in the chi-square and multi-level analysis. Is there a reason why?

Response: This variable was dropped and not utilised throughout the study. Hence, its absence from the Tables. 

3. While the literature on tetanus immunization uptake have been highlighted, I suggest that you could make use of a theoretical background or framework which provides a logical relationship between the variables and also help establish apriori expectations. The framework would provide a solid rationale for the work. For instance the socio-ecological model can be useful. It will enable you look at the multiple levels of influence on health behaviors or variables and their outcomes. This can also help you delineate the individual level variables and contextual variables properly

Response: We have provided a theoretical model that informed the selection of the variables included in the study.

4. Lastly, the paper will benefit from some proof reading. For instance in the discussion section ‘update’ should read ‘uptake’ and ‘them themselves’ should read ‘ themselves’ .

Response: Thank you. We have thoroughly proofread the manuscript to correct grammatical and typographical errors.

---

## [Editor Report · Decision Letter 2]

8 Dec 2023

Prevalence and correlates of tetanus toxoid uptake among women in sub-Saharan Africa: Multilevel analysis of demographic and health survey data

PONE-D-23-03453R2

Dear Dr. Dowou,

We’re pleased to inform you that your manuscript has been judged scientifically suitable for publication and will be formally accepted for publication once it meets all outstanding technical requirements.

Kind regards,

Frank T. Spradley

Academic Editor

PLOS ONE

---

## [Editor Report · Acceptance letter]

15 Dec 2023

PONE-D-23-03453R2 

PLOS ONE

Dear Dr. Dowou, 

I'm pleased to inform you that your manuscript has been deemed suitable for publication in PLOS ONE. Congratulations! Your manuscript is now being handed over to our production team.

Kind regards, 

on behalf of

Dr. Frank T. Spradley 

Academic Editor

PLOS ONE